# Superelasticity of a photo-actuating chiral salicylideneamine crystal

Takuya Taniguchi [1✉], Kazuki Ishizaki[2], Daisuke Takagi[3], Kazuki Nishimura[4], Hiroki Shigemune [4], Masahiro Kuramochi[5], Yuji C. Sasaki [6], Hideko Koshima [7] & Toru Asahi[2,3,7]

Superelasticity is a type of elastic response to an applied external force, caused by a phase transformation. Actuation of materials is also an elastic response to external stimuli such as light and heat. Although both superelasticity and actuation are deformations resulting from stimulus-induced stress, there is a phenomenological difference between the two with respect to whether force is an input or an output. Here, we report that a molecular crystal manifests superelasticity during photo-actuation under light irradiation. The crystal exhibits stepwise twisted actuation due to two effects, photoisomerization and photo-triggered phase transition, and the actuation behavior is simulated based on a dynamic multi-layer model. The simulation, in turn, reveals how the photoisomerization and phase transition progress in the crystal, indicating superelasticity induced by modest stress due to the formation of photo-products. This work provides not only a successful simulation of stepwise twisted actuation, but also to the best of our knowledge the first indication of superelasticity induced by light.

[1] Center for Data Science, Waseda University, 1-6-1 Nishiwaseda, Shinjuku-ku, Tokyo 169-8050, Japan. [2] Department of Advanced Science and Engineering, Graduate School of Advanced Science and Engineering, Waseda University, 3-4-1 Okubo, Shinjuku-Ku, Tokyo 169-8555, Japan. [3] Department of Life Science and Medical Bioscience, Graduate School of Advanced Science and Engineering, Waseda University, 3-4-1 Okubo, Shinjuku-Ku, Tokyo 169-8555, Japan. [4] Department of Electrical Engineering, Shibaura Institute of Technology, 3-7-5 Toyosu, Koto-Ku, Tokyo 135-8548, Japan. [5] Department of Materials Science and Engineering, Graduate School of Science and Engineering, Ibaraki University, 4-12-1 Naka-Narusawa-cho, Hitachi-Shi, Ibaraki 316-8511, Japan. [6] Graduate School of Frontier Sciences, The University of Tokyo, Kashiwa 277-8561, Japan. [7] Research Organization for Nano & Life Innovation, Waseda University, 513 Wasedatsurumaki-cho, Shinjuku-Ku, Tokyo 162-0041, Japan. ✉email: takuya.taniguchi@aoni.waseda.jp

The relationship between stress and strain is fundamental in material mechanics to characterize the deformation behaviors of materials and structured objects. When an external force is applied to an object, it deforms due to elastic and/or plastic responses followed by fracture, depending on the mechanical properties. In some cases, superelasticity, a pseudo-elastic deformation caused by a phase transition, appears due to applied force. Although the manifestation of superelasticity is famous in shape memory alloys[1,2], molecular crystals have recently shown superelasticity despite their rigid and fragile appearance[3–5]. The responses of molecular crystals to the applied force are being researched with substantial curiosity as crystal adaptronics[6].

As another elastic response, when stress is induced to material by external stimuli such as light, heat, and electromagnetic fields, the material exerts actuation. Actuation materials can output deformation and force. Whereas conventional actuators consisting of hard materials have been utilized in many applications, organic smart actuators have attracted attention recently owing to their softness and flexibility[7–10]. Promising applications of organic actuators include soft robots, flexible devices, wearable electronics, and micromechanical electrical systems[11–13]. Although polymeric materials have been the mainstay of soft smart actuators[14,15], molecular crystals have shown actuation behaviors induced by light and heat, as summarized in several reviews[16–19].

Molecular crystals have displayed superelasticity and actuation, and there is a phenomenological difference between them whether force is an input or output. In either case, the resultant deformation will be elongation, shear deformation, bending, twist, and so on depending on the anisotropy of mechanical property and induced stress. From the actuation point of view, bending is a basic actuation that affords large displacements from a small strain. The simplest way to induce bending is to generate a stress gradient in a material. When a thermal phase transition occurs in a molecular crystal due to a temperature change, the crystal bends because of the coexistence of two crystal phases[20–23]. Photo-bending is also produced by a gradient of photoproducts due to photoisomerization[24–26] and is approximated by Stoney's and Timoshenko's bilayer models assuming a photoisomerization layer[27–29]. Although crystals most commonly grow in a straight shape, it has been found that many kinds of

crystals naturally and potentially grow in twisted shapes by several mechanisms such as screw dislocation and Eshelby twist[30]. In the case of photo-actuation, anisotropic stress by photoproducts sometimes causes a twisting deformation[31–34], which is much rarer than bending. The twisted actuation has been, in some cases, analyzed successfully by analytical models[34]. Thus, the actuation of bending or twisting can be analyzed by constructed models to some extent, and once the model parameters are adjusted, the actuation behavior becomes predictable and controllable.

However, when bending and twisting occur simultaneously owing to two different effects, the actuation behavior becomes more complicated. In fact, a molecular crystal of the compound (S)-N-3,5-di-tert-butylsalicylidene-1-(1-naphthyl)ethylamine [enol-(S)-1] (Fig. 1a), in which photoisomerization and photo-triggered phase transition (PtPT) occur simultaneously, exhibited anomalous stepwise twisted actuation[35]. Furthermore, PtPT is considered to be a phase transition that occurs due to the accumulation of stress by a small number of photoproducts (Fig. 1b); thus, the proposed mechanism is similar to that of superelasticity in the point that stress causes phase transformation and de-stress causes the reverse transition. In spite of the findings of the previous study, further analysis of the conditions and mechanism of PtPT is required to conclude the superelasticity at PtPT.

This work shows that the molecular crystal of the enol-(S)-1 compound exhibits superelasticity during photo-actuation. The elastic property and actuation performance were characterized, and then a dynamic multi-layer model was introduced to simulate the stepwise twisted actuation using finite element analysis (FEA). The simulation, in turn, revealed how photoisomerization and PtPT progressed in the crystal and indicated superelasticity induced by internal stress due to photoproducts. This work not only successfully simulates stepwise twisted actuation but also provides, to the best of our knowledge, the first indication of superelasticity induced by light.

## Results and discussion

**Elastic properties.** Typically, thin plate-like crystals of enol-(S)-1 were obtained with the (001)/(00$\bar{1}$) face as the top surface, (100)/

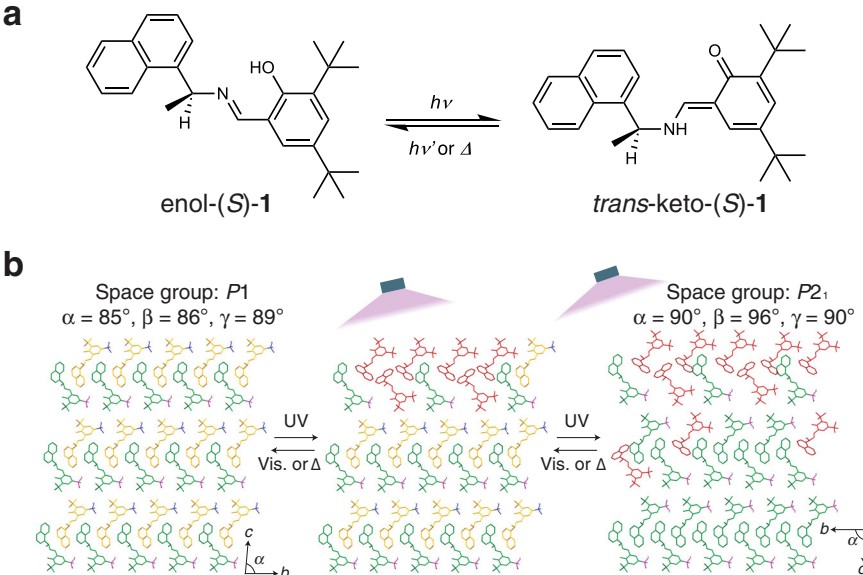

**Fig. 1 Photo-triggered phase transition (PtPT). a** Photoisomerization of enol to *trans*-keto form. **b** Proposed mechanism of the PtPT and the measured lattice constants[35]. Enol-(S)-**1** molecules in green and yellow reflect $Z' = 2$, and *trans*-keto-(S)-**1** is shown in red.

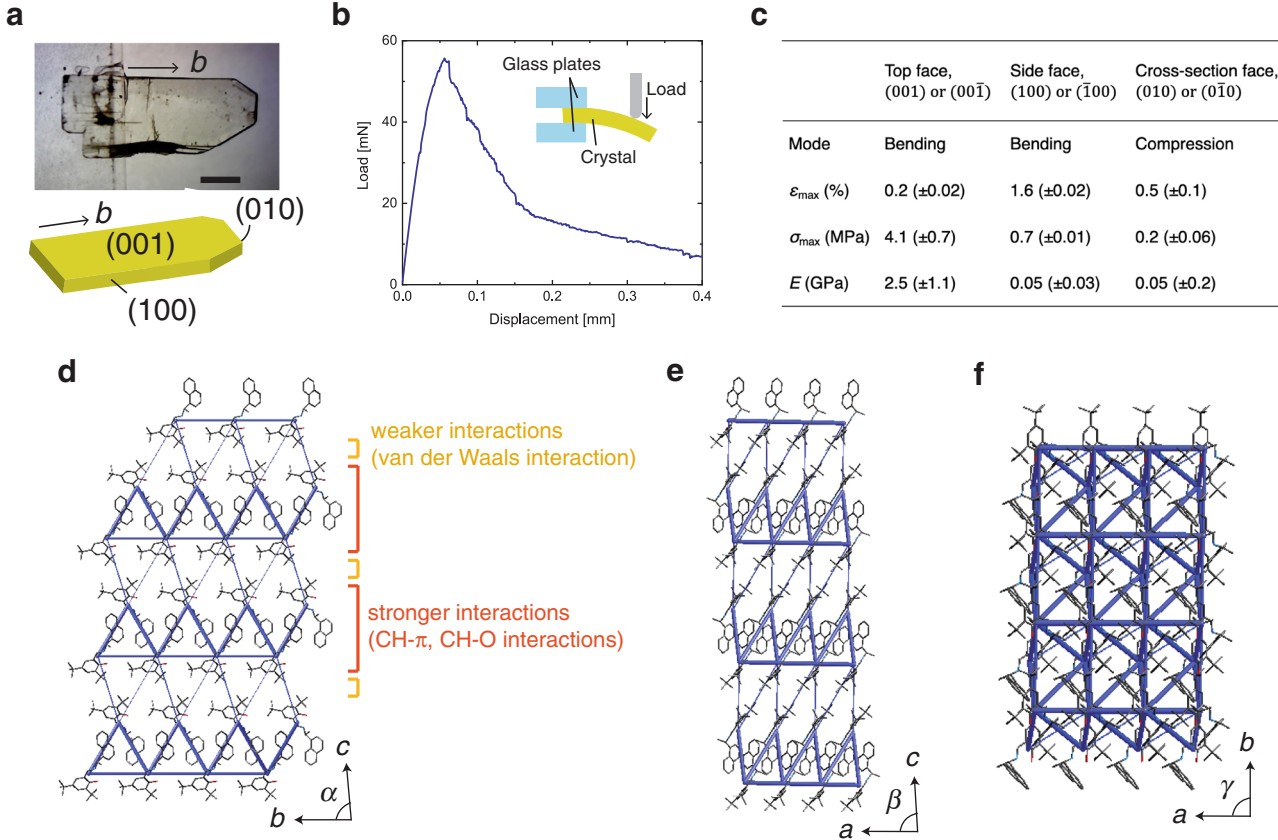

**Fig. 2 Crystal structures and Young's modulus of enol-(S)-1 crystals. a** Photograph and illustration of crystal shape and face index. Scale bar is 1 mm. **b** Typical load–displacement curve of an enol-(S)-1 crystal-loaded onto the (001)/(00$\bar{1}$) top face. Inset is a side view of the measurement setup. **c** Table of elastic responses when loaded on top, side, and cross-section faces. **d–f** Molecular packing and energy framework, viewed from the ($\bar{1}$00) side face (**d**), (010) cross-section face (**e**), and (00$\bar{1}$) top face (**f**).

($\bar{1}$00) face as the side face and (010)/(0$\bar{1}$0) face as the cross-section face along the longitudinal direction of the b-axis (Fig. 2a). The elastic modulus was determined from load-displacement curves by beam bending or compression depending on the loaded crystal face. When a load was applied to the (001)/(00$\bar{1}$) surface of an enol-(S)-1 crystal, elastic response was observed up to approximately 40 mN (Fig. 2b). The Young's modulus when loaded on the (001)/(00$\bar{1}$) face was calculated to be 2.5(±1.1) GPa from the results of three samples (Fig. 2c and Supplementary Note 1), and the elastic response was repeated without fracture at least 10 times (Supplementary Fig. 1). Based on the elastic response, the maximum strain $\varepsilon_{max}$ and stress $\sigma_{max}$ were calculated to be 0.2(±0.02)% and 4.1(±0.7) MPa, respectively (Fig. 2c). In contrast to the elastic property loaded on the top face, when a load was applied to the (100)/($\bar{1}$00) side face and (010)/(0$\bar{1}$0) cross-section face, Young's moduli were much smaller (Fig. 2c and Supplementary Fig. 2). Smaller $E$ and $\sigma_{max}$ reflect the anisotropy of the crystal structure and suggest the existence of weaker intermolecular interactions on the (100)/($\bar{1}$00) and (010)/(0$\bar{1}$0) faces compared to the interactions on the (001)/(00$\bar{1}$) face.

The crystal structure of enol-(S)-1, which has been determined in the previous research[35], forms two relatively stronger intermolecular interactions: CH–π interactions and CH–O hydrogen bonds (Fig. 2d). CH–π interactions are formed mainly by the face-to-edge stacking of naphthyl rings along the b-axis, and CH–O interactions are formed between the OH group of a molecule and the tert-butyl group of the adjacent molecule along the b-axis. The stronger interaction layer is stacked along the c-

axis through a weaker interaction layer by van der Waals interactions between the bulky tert-butyl substituents. The energy framework was calculated to evaluate the strength of these intermolecular interactions, and it was confirmed that stronger and weaker interaction layers are arranged alternately along the c-axis on the (100) face (Fig. 2d). The view from the (010) face also represents an alternating arrangement of stronger and weaker interaction layers, indicating that these layers extend in two dimensions along the a- and b-axes (Fig. 2e). The view from the (001) face, in contrast, shows only stronger interactions due to the layer-by-layer stacking (Fig. 2f).

This anisotropy of the crystal structure is consistent with the observed elastic moduli. There are stronger and weaker interaction layers alternately on the (100) and (010) faces, and thus applying load on these faces should result in cleaving the weaker intermolecular interactions by relatively smaller stress. The enol-(S)-1 crystals loaded on these faces displayed this cleavage behavior (Supplementary Fig. 3). In contrast, loading on the (001) face suppresses such cleavage due to stronger intermolecular interactions, resulting in a larger elastic modulus.

**Actuation performance.** The enol-(S)-1 crystal deforms when irradiated by UV light (365 nm) on the (001) or (00$\bar{1}$) face, and the deformation can be divided into three steps (Fig. 3a). The first step involves simple bending toward the light source due to photoisomerization. The second step is twisted bending due to the progression of PtPT, as shown by the most twisted shape. The third step is simply bending toward the light source due to photoisomerization after completion of PtPT.

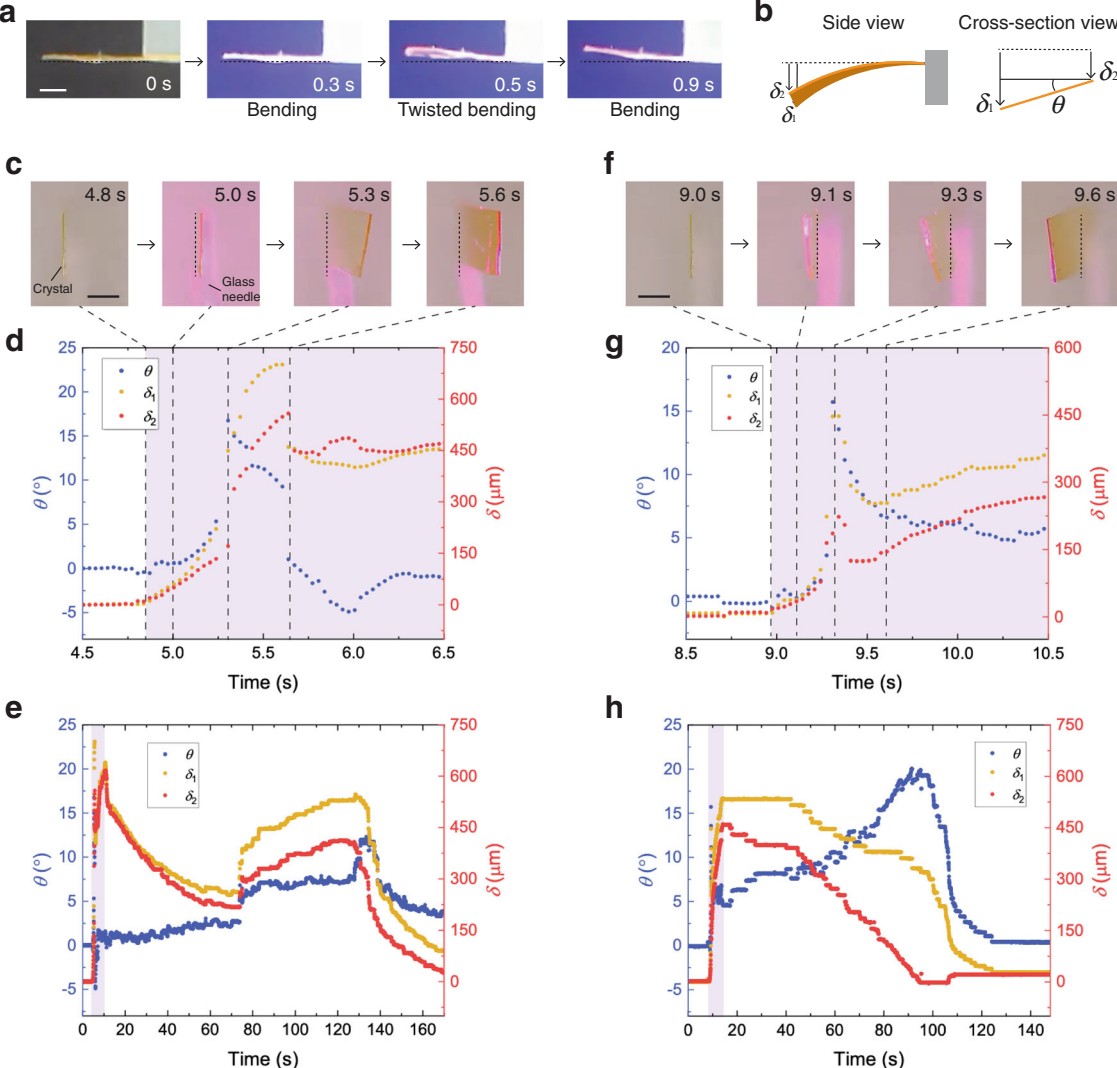

**Fig. 3 Deformation behavior upon photoirradiation. a** Photographs of typical deformation of an enol-(S)-**1** crystal, fixed on a glass plate, irradiated by UV light (365 nm). Scale bar is 1 mm. **b** Definition of torsion angle $\theta$ and displacements $\delta_1$ and $\delta_2$ ($\delta_{1,max} > \delta_{2,max}$). Dotted lines are the assumed initial position. **c–e** Cross-section view of an enol-(S)-**1** crystal, fixed on a glass needle, irradiated on the (001) face (**c**) and time-series data of $\theta$, $\delta_1$, and $\delta_2$ at the initial irradiation (**d**) and full scale (**e**). **f–h** Cross-section view of the enol-(S)-**1** irradiated on the (00$\bar{1}$) face (**f**) and time series data of $\theta$, $\delta_1$, and $\delta_2$ at the initial irradiation (**g**) and full scale (**h**). Scale bars in **c** and **f** are 0.5 mm. The regions highlighted in purple represent under UV light at 180 mW cm$^{-2}$.

Such stepwise actuation can be quantified by the torsion angle $\theta$ and displacements $\delta_1$ and $\delta_2$ at two edges, where $\delta_{1,max} > \delta_{2,max}$ by definition (Fig. 3b). When the (001) face of an enol-(S)-**1** crystal (4.0 mm, 0.94 mm, 48 µm) was irradiated by UV light at 180 mW cm$^{-2}$, the crystal viewed from the cross-section direction exhibited the stepwise motion as mentioned above, with a left-handed twist (Fig. 3c and Supplementary Movie 1). Quantified $\delta_1$ and $\delta_2$ were almost the same during the initial illumination (4.8–5.0 s), and then $\delta_1$ increased more than $\delta_2$ due to twisted bending by the propagation of PtPT (5.0–5.3 s) (Fig. 3d). After the most twisted position at 5.3 s, the torsion angle decreased and then suddenly resolved due to the completion of PtPT (5.6 s), followed by a gradual increase in displacements at almost the same values (Fig. 3d).

This stepwise deformation appeared within a few seconds upon photo-illumination, and after the cessation of illumination, stepwise relaxation of simple unbending (15–75 s) → twisted unbending (75–140 s) → simple unbending (140–170 s) was observed because of the back-isomerization reaction and the reverse propagation of the phase transition (Fig. 3e). Considering

that the half-life of the *trans*-keto form is 86.2 s at 20 °C[35], twisted unbending started when almost half of the *trans*-keto form returned to the enol form, and the twisted shape disappeared when most of the *trans*-keto form returned to the enol form.

In the case of illumination on the (00$\bar{1}$) face, a similar stepwise motion with a right-handed twist was observed (Fig. 3f and Supplementary Movie 2). The anomalous peak of displacements and torsion angle also appeared due to the propagation of PtPT, while some degree of $\theta$ remained probably due to residual stress even after the completion of PtPT (Fig. 3g). After stopping the light irradiation, stepwise unbending similar to the case of the (001) face was observed, and twisted unbending appeared noticeably at 60–110 s (Fig. 3h). Thus, stepwise twisted actuation and relaxation occurred under and after UV light irradiation on the (001) and (00$\bar{1}$) faces, although there were some differences in the time-series behavior, probably due to residual stress or other unidentified effects. In addition, twist-handedness should be dependent on the irradiated face of the chiral enol-(S)-**1** crystal, as confirmed by another face-indexed crystal (Supplementary Fig. 4).

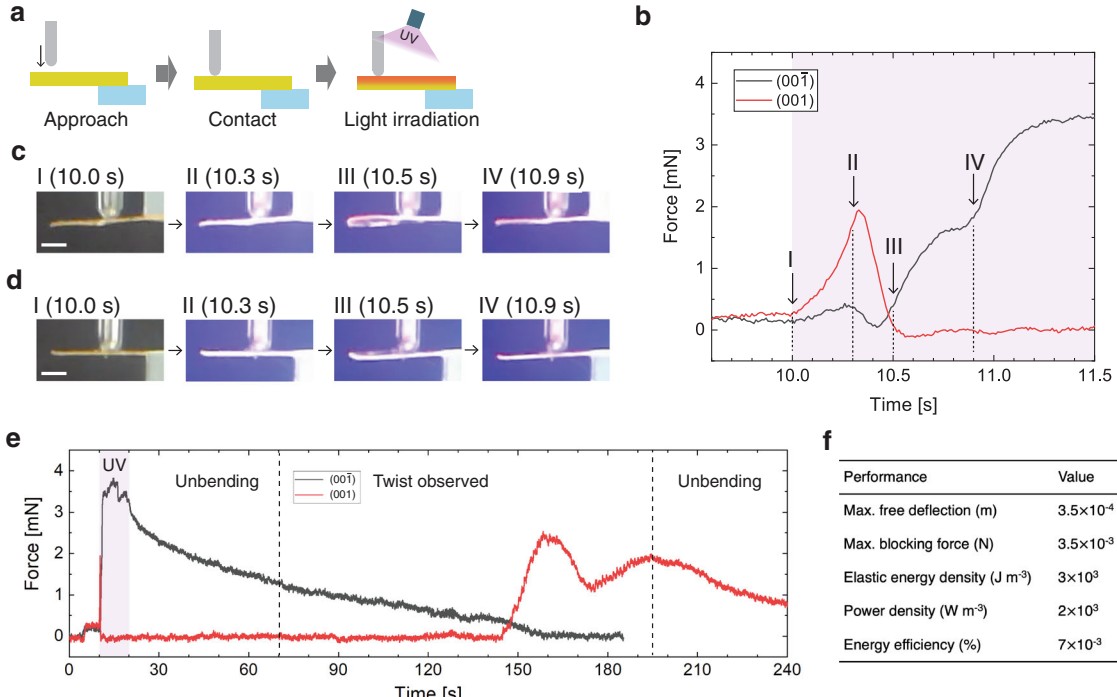

**Fig. 4 Time dependence of blocking force upon light irradiation. a** Measurement procedure of blocking force. **b** Initial process of force generation upon light irradiation on the (001) and (00$\bar{1}$) faces. **c, d** Photographs of the crystal with load upon light irradiation on the (00$\bar{1}$) (**c**) and (001) (**d**) face. Each picture under loading corresponds to the states I–IV in panel (**b**), simultaneously observed. Scale bars are 1 mm. **e** Time dependence of blocking force after the cessation of light irradiation. The period of twist existence was confirmed from recorded movies. **f** Metrics of actuation performance of the crystal used in this figure.

Force is another fundamental output for actuation although there have been only a few reports on the force generated by photo-actuating molecular crystals[28,36–38]. The blocking force of an enol-(S)-**1** crystal (3.8 mm, 1.8 mm, 55 μm) was measured by the previously reported procedure (Fig. 4a)[38], with light irradiation conducted for 10 s (10–20 s). When the (00$\bar{1}$) face was irradiated by UV light at 180 mW cm$^{-2}$, the blocking force increased 0.4 mN at 10.3 s due to the prevention of simple bending (Fig. 4b, c and Supplementary Movie 3). Then, through a slight decline by initiating the PtPT, blocking force again increased at 10.5 s, where the most twisted shape of right-handedness was observed (Fig. 4b, c). The force increase and twisted shape continued until 10.9 s, after which the force increased again owing to the prevention of simple bending after the completion of PtPT, reaching a maximum value of 3.5 mN at the steady-state (Fig. 4b).

In the case of irradiation of the (001) face, blocking force increased 1.7 mN at 10.3 s by the prevention of simple bending, and then decreased due to the PtPT, where the most twisted shape of left-handedness was observed at 10.5 s (Fig. 4b, d and Supplementary Movie 4). After this state, the blocking force was almost zero because the crystal surface detached from the jig. The detachment may be caused by the deformation due to the PtPT, and in fact, the crystal tip at 10.9 s was below the initial position at 10.0 s (Fig. 4d). Although blocking force was not detected after the detachment, twisted bending continued up to 10.9 s, and then slight simple bending toward the light source was observed during photo-illumination (Supplementary Movie 4).

After the cessation of the light irradiation at 20 s, the force of the crystal irradiated on the (001) and (00$\bar{1}$) faces also behaved differently (Fig. 4e), but originated from the same stepwise relaxation: simple unbending (20–70 s), twisted unbending (70–195 s), and subsequent simple unbending (195 s). The force on the (00$\bar{1}$) face continuously decreased after light cessation, and

slight torsion was observed after 70 s. The force decrease continued until 160 s with a twisted shape and became zero, possibly due to detachment from the jig. Thereafter, the force was not detected while the crystal shape was still slightly twisted (Supplementary Movie 3). The force measured on the (001) face was almost zero until 145 s because of detachment from the jig, but then increased by re-contact. The blocking force reached peaks at 160 and 195 s with a slightly twisted shape, and then continuously decreased without noticeable torsion (Fig. 4e and Supplementary Movie 4).

Based on the blocking force and free deformation of the enol-(S)-**1** crystal shown in Fig. 4, the actuation performance was evaluated (Fig. 4f and Supplementary Note 2). The enol-(S)-**1** crystal exhibited a maximum displacement of 350 μm and maximum force of 3.5 mN at the steady-state under UV light. The value of the maximum force is comparable to similar salicylide-neamine crystals and some azobenzene crystals[28,38], and this suggests that the output force does not decrease as much, even though the PtPT consumes some energy. The elastic energy density and power density were also calculated to be 3 kJ m$^{-3}$ and 2 kW m$^{-3}$, respectively, based on the region of simple bending because twisted bending is difficult to consider in these calculations. These values are on the high side for photo-actuating molecular crystals[28]. In addition, the energy conversion efficiency was estimated to be 0.007%, which is higher than that of most light-driven polymeric actuators and comparable to photo-bending azobenzene crystals[28].

**Manifestation condition of PtPT**. Photoisomerization is the origin of the manifestation of PtPT, and thus the characterization of the conversion ratio of photoisomerization is important. For this purpose, Fourier-transform infrared (FT-IR) measurements were employed because there is a difference between the enol and

*trans*-keto forms due to intramolecular proton transfer. Theoretical calculations revealed that the IR spectra of the enol form had O–H oscillations near 3000 cm$^{-1}$, whereas that of the *trans*-keto form did not (Supplementary Fig. 5a). The conversion ratio from enol to *trans*-keto form can be calculated based on this difference, and the FT-IR spectra of the enol-(*S*)-**1** powder before and under UV irradiation of 180 mW cm$^{-2}$ showed a slight decrease in the absorption assigned to O–H stretching (Supplementary Fig. 5b), suggesting that the photoconversion ratio should be 5% in the steady-state. This value is consistent with the previous research that the conversion ratio of photoisomerization was insufficient for an X-ray structure determination[35] and comparable with the conversion ratio of similar photochromic crystals[24,34]. This estimation roughly indicates that the PtPT starts when the ratio of photoproducts is nearly 1% in the crystal based on the time-series behaviors shown in Figs. 3 and 4.

Furthermore, the dependence of the light intensity on the PtPT was evaluated. When the crystal in Fig. 4 was irradiated at several light intensities (20–360 mW cm$^{-2}$), the occurrence and completion of PtPT became faster depending on the increase of light intensity (Supplementary Fig. 6). To consider the photothermal effect on this behavior, the surface temperature was monitored using a thermography camera, and the surface temperature was increased to less than 40 °C, which is the thermal transition temperature of the enol-(*S*)-**1** crystal, at weaker intensities up to 180 mW cm$^{-2}$ (Supplementary Fig. 7). Even when the surface temperature increased to more than 40 °C, the PtPT appeared before the temperature increased to the thermal transition temperature. In contrast, another crystal with a larger volume (3.3 × 3.3 × 0.27 mm$^3$) exhibited twisted shape and force peaks corresponding with the timing of temperature rise to almost 40 °C (Supplementary Fig. 8). These results suggest that smaller crystals exhibit phase transition due to the effect of photoproducts (i.e., PtPT), whereas larger crystals exhibit phase transition by the heating effect (*i.e.*, thermal transition), and the border is likely in the volume range of 2–2.5 mm$^3$ (Supplementary Fig. 9).

**Simulation of actuation and indication of superelasticity.** Because twist-bent deformation is caused by PtPT, which occurs during the initial photo-process in relatively smaller crystals, it is important to reproduce the actuation behavior by simulating the material mechanics for controlling the actuation behavior and understanding the actuation mechanism. For this purpose, we used FEA, which is commonly used in the field of material mechanics[39,40]. To simulate the deformation behavior using FEA, it is necessary to construct a simpler model of actuation. As shown in Fig. 3, the enol-(*S*)-**1** crystal exhibits stepwise actuation: simple bending, twisted bending, and subsequent simple bending. As such, we can consider a dynamic multi-layer model, in which the photoisomerization layer is $h_1$, the PtPT layer is $h_2$, and the remaining layer is $h_3$ where crystal thickness is $h_0$ (Fig. 5a). It is also assumed that $h_1$ and $h_2$ are generated from the top surface, considering the light irradiation from the upper surface. In addition, $h_1$ can be set much smaller than $h_0$ because photoproducts are generated at most 5%, and the maximum of $h_2$ can be set equal to $h_0$. Based on this model, FEA was performed with the primary goal of reproducing both simple bending and twisted bending, and once this was achieved, we investigated the thickness dependence of $h_1$ and $h_2$ on the deformation to clarify how photoisomerization and PtPT progress in the crystal.

FEA first requires material geometry and physical properties. The material geometry was defined as 4.0 mm in length, 0.96 mm in width, and 50 μm in thickness ($h_0$), which is almost the same size as the crystal in Fig. 3. The physical properties and stimulus parameters are summarized in Supplementary Fig. 10. The length

contraction (−0.8%) of $h_1$ and shear deformation (Δ1°) with slight contraction (–0.05%) of $h_2$ were determined by referring to the results of X-ray diffraction analysis in a previous study[35] (Fig. 5b). Here, length contraction and shear deformation were created by thermal contraction and piezoelectric effect, respectively, because photo-effects cannot be incorporated directly into FEA. Then, the static deformation of the multi-layer model was created by changing the thicknesses of $h_1$–$h_3$ so that simple bending in state II, twisted bending in state III, and simple bending in state IV were successfully reproduced (Fig. 5c). The maximum displacement $\delta$ and torsion angle $\theta$ can be evaluated using the definition in Fig. 3b. In addition, the difference in twist handedness depending on the irradiated face was also reproduced successfully (Supplementary Fig. 11).

After the assumed model reproduced the observed deformation, the thickness dependence of $h_1$ and $h_2$ on deformation was further investigated. The values of $h_1$ and $h_2$ were varied in the range of 0–5 and 0–50 μm, respectively, and then polynomial regression was performed to describe the response surface of the torsion angle and maximum displacement (Fig. 5d, e). This response function depends on $h_1$ and $h_2$, but does not include the time factor. Thus, by assuming $h_1$ is proportional to the progress of the photoisomerization reaction, $h_1$ can be expressed as

$$h_1(t) = h_{1,\max}\left\{1 - \exp\left(-\frac{t}{\tau_{1p}}\right)\right\}(t \geq 0) \quad (1)$$

based on first-order chemical kinetics. Here, $h_{1,\max}$ is the maximum depth of the photoisomerization layer, $t$ is the light irradiation time, and $\tau_{1p}$ is the time constant. The progress of $h_2$ also depends on photoisomerization, and the effect of photoproducts causes a phase transition. Thus, it is assumed that $h_2$ is expressed by a similar exponential function with an independent time constant and some delay $t_{delay}$ from $t = 0$

$$h_2(t) = h_{2,\max}\left\{1 - \exp\left(-\frac{t}{\tau_{2p}}\right)\right\}\left(t \geq t_{delay}\right) \quad (2)$$

where $h_{2,\max}$ is the maximum depth of PtPT, and $\tau_{2p}$ is the time constant. Because $h_{2,\max}$ can be $h_0$, the other parameters $h_{1,\max}$, $\tau_{1p}$, $\tau_{2p}$, and $t_{delay}$ are to be optimized. In the analysis of the relaxation process, time factors $h_1$ and $h_2$ were introduced based on the assumption that the photo-process, and thus $h_1$ and $h_2$ can be represented as

$$h_1(t) = h_{1,\max}\exp\left(-\frac{t}{\tau_{1r}}\right)(t \geq 0) \quad (3)$$

$$h_2(t) = h_{2,\max}\exp\left(-\frac{t}{\tau_{2r}}\right)\left(t \geq t_{delay}\right) \quad (4)$$

where $\tau_{1r}$ and $\tau_{2r}$ are the relaxation time constants. Here, the parameters $\tau_{1r}$, $\tau_{2r}$, and $t_{delay}$ are to be optimized, as the others have been determined in the photo-process.

As a result of parameter optimization (Supplementary Fig. 12), the FEA-based simulations were well fitted with the observed torsion angle and displacement of the crystal in both photo and relaxation processes (Fig. 5f–i), although there is some misfit in displacement. This simulation outperformed classical Stoney's bimorph model for predicting the displacement behavior (Fig. 5h, i). The experimental results on (00$\bar{1}$) were also simulated through parameter optimization, and twisted actuation was successfully simulated, although there was a larger misfit than that for the (001) face (Supplementary Figs. 13 and 14). Such comparison of experiment and simulation was performed using other enol-(*S*)-**1** crystals, validating the assumed model (Supplementary Fig. 15).

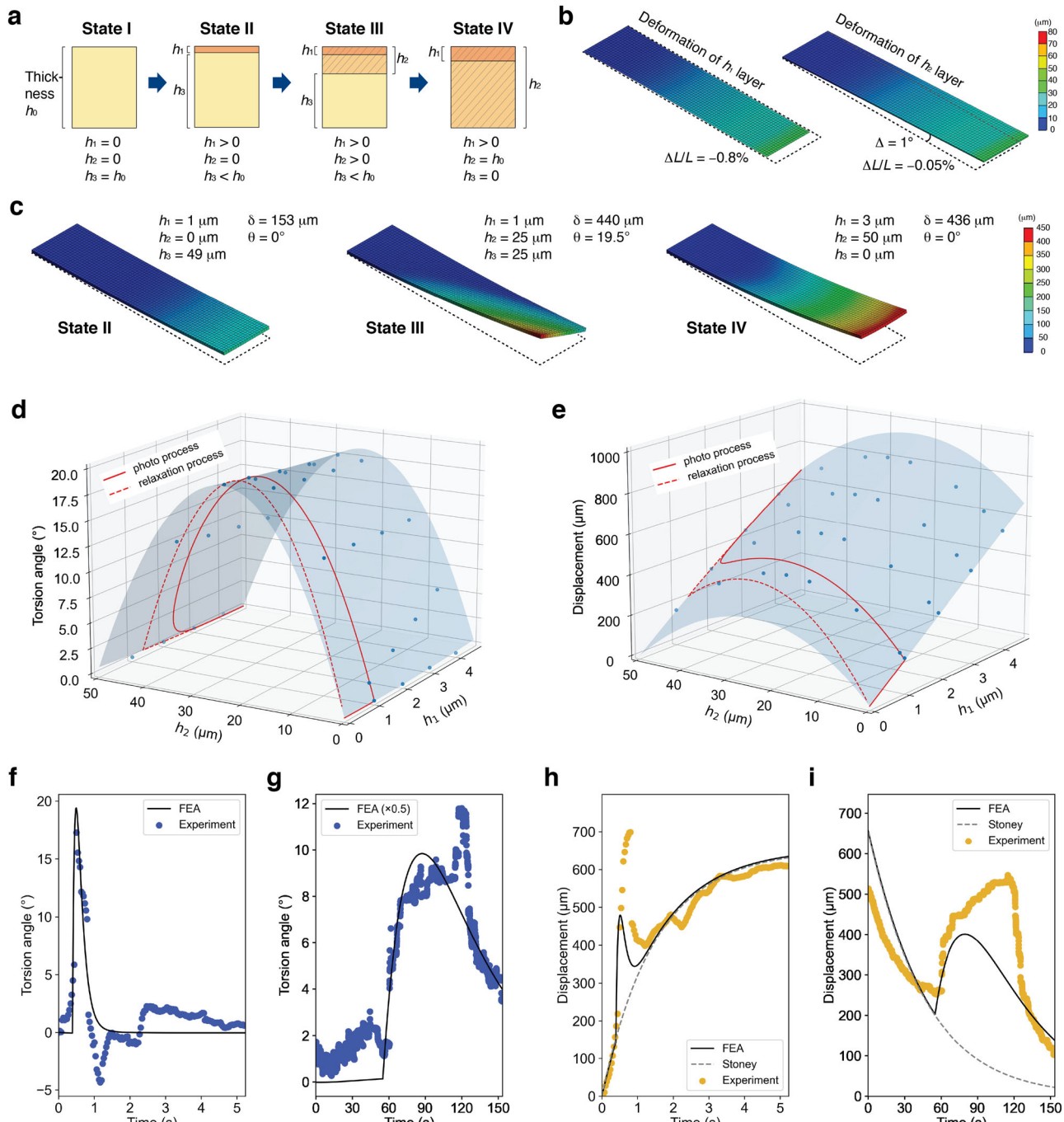

**Fig. 5 Simulation of crystal deformation by finite element analysis (FEA). a** Dynamic multi-layer model in which four states I–IV were categorized by the thicknesses of $h_1$ and $h_2$. **b** Independent deformations of $h_1$ and $h_2$ layers, which imitate the effects of photoisomerization and photo-triggered phase transition, respectively. The original dimensions of the plate object are 4.0 mm in length and 0.94 mm in width. Deformation is enhanced 6.4 times because raw deformation is much smaller than the object size. **c** Simulated typical deformation at three states: II–IV. **d, e** Simulated dependence of torsion angle (**d**) and maximum displacement (**e**) on the thicknesses of $h_1$ and $h_2$ layers. Blue dots are the simulated data points, and the response surfaces are drawn by a polynomial function fitted to the simulated points. Red lines are the estimated route that reproduces the observed torsion angle and displacement. **f, g** Comparison of the simulation and the torsion angle in the photo-process (**f**) and relaxation process (**g**). In **g**, the simulation was scaled by half for fitting. **h, i** Comparison of the simulation and the displacement in the photo-process (**h**) and relaxation process (**i**). The experimental results come from Fig. 3e, and time was rescaled for each process.

The optimized results, in turn, address how the $h_1$ and $h_2$ layers progressed in the crystal (red lines in Fig. 5d, e). In the photo-process, the value of $h_{1,max}$ was 4.7 μm, and the $h_2$ layer started at $h_1 \sim 1$ μm with a delay time of 0.38 s and progressed 10 times faster than the $h_1$ layer, producing the maximum torsion angle

around $h_2 = h_0/2$. Then, the $h_2$ layer reached $h_0 = 50$ μm, and subsequently, the $h_1$ layer increased up to $h_{1,max}$. In the relaxation process, only $h_1$ decreased initially, and then $h_2$ started to decrease at $h_1 \sim 1.5$ μm with a delay time of 55 s. The relaxation speeds of $h_1$ and $h_2$ were the same, based on the fitting. It is

notable that there is a hysteresis between the photo and relaxation processes. Considering that photoproducts are the origin of stress to induce PtPT and that de-stress of photoproducts leads to the reverse transition, this hysteresis should be the same representation of conventional superelasticity, in which hysteresis of the stress-strain curve is generally observed. Therefore, it can be said that superelasticity appears at the PtPT owing to the stress caused by photoproducts.

Indeed, phase boundary and slight shear deformation were observed during PtPT and reverse transition under a polarized microscope (Supplementary Fig. 16 and Supplementary Movie 5). The phase boundary was generated parallel to (001) face, to which UV irradiation was conducted. This appearance of phase boundary is common with superelasticity, and the shear deformation of enol-(S)-**1** crystal was much smaller than those of other superelastic crystals[3–5]. Furthermore, the anisotropy of PtPT expression was investigated (Supplementary Fig. 17). UV irradiation to (010) face led to the partial generation of phase boundary, while light irradiation to (100) face did not cause such boundary. This observation suggests that light irradiation to (001) face is the most effective to induce PtPT probably due to the anisotropy of crystal structure, photo-reactivity, and a large area of the face. In all observations, we did not detect any local melt, which would affect the PtPT and deformation behavior (Supplementary Fig. 17).

**Structure dynamics and proposed mechanism.** To investigate the manifestation mechanism of the PtPT, the dynamics of the crystal structure were evaluated using diffracted X-ray blinking (DXB). This technique measures time-resolved X-ray diffraction images, and then analyzes the time-series intensity at each pixel of specific diffraction by an autocorrelation function (ACF), and eventually clarifies the dynamic changes in the crystal structure[41,42] (Supplementary Fig. 18 and Supplementary Note 3). By means of DXB, the dynamics of diffraction from the (003) plane were evaluated before and under UV light (Fig. 6a–c). The decay constant, which should reflect the speed of lattice fluctuation of the (003) plane, was distributed mostly at smaller values, although some larger values were also observed (Fig. 6b). The boxplot clearly shows that the larger decay constants decreased in the initial few seconds of photoirradiation, indicating that lattice fluctuation was suppressed, probably by the generated stress during PtPT. The stress may include the effects of both the photoproducts and the phase boundary. After prolonged irradiation for 50 s, the lattice dynamics became similar to that before light irradiation, suggesting that large stress was resolved after the completion of PtPT. Considering that the number of photoproducts increases under light up to a steady-state, the distinct suppression of lattice fluctuation may be influenced by the stress from the phase boundary during the propagation of PtPT.

This dynamics change can be interpreted as shown in Fig. 6d. The dynamics of the (003) planes are regulated by both stronger interactions (CH–π and CH–O) and weaker van der Waals interactions (Fig. 6d). Under light irradiation, the photoproducts generate compressive stress along the *b*-axis through stress induction to adjacent molecules. Based on the strength of intermolecular interactions, compressive stress transfers effectively owing to the stronger interactions of CH–π and CH–O interactions, whereas stress transfer along the *c*-axis is less effective because of the weaker van der Waals interactions. Once a certain ratio (~1%) of photoproducts is generated, PtPT begins owing to the small energy barrier ($\Delta H = 0.2$ kJ mol$^{-1}$)[35]. The PtPT generates a phase boundary between the raw phase parallel to the (001) face, and the stress caused by the phase boundary

influences the dynamics of the (003) plane. DXB in the initial photoirradiation should capture this change in the crystal structure dynamics. When PtPT is completed, the phase boundary disappears, and the stress remaining comes from the photoproducts. Based on the results of DXB, the stress from the photoproducts may have a weaker influence on the lattice dynamics of the (003) plane than that of the phase boundary.

The modest stress from the photoproducts may be crucial for the superelastic deformation at PtPT. In other molecular crystals, superelasticity has been induced by applying a shear force, and the estimated shear stress was much larger than that of the PtPT observed here. This suggests that it is difficult for the enol-(S)-**1** crystal to manifest superelasticity by an external force, and we did not observe such an indication. This may indicate that there is a difference between physically applied force and chemically induced stress.

In summary, this study provides the first indication of superelasticity during actuation by light. Stepwise twisted actuation was successfully simulated by a dynamic multi-layer model, and the simulation suggested the superelastic behavior of the PtPT in the enol-(S)-**1** crystal. Although superelasticity is thought to be a response to external force, this study extends the limitation by the combined system of superelasticity and actuation in a single crystal. These findings should contribute to the development of novel mechanical materials.

## Methods

**Material preparation.** An enol-(S)-**1** compound was synthesized using microwaves (Monowave 300, Anton Paar, Graz, Austria), according to the literature[43]. Plate-like single crystals of enol-(S)-**1** were obtained by evaporation of 2-propanol or acetonitrile solution at ambient temperature.

**Calculation of energy framework.** The crystal structures of enol-(S)-**1** have previously been reported[35]. The energy framework was drawn based on the intermolecular interaction energies, which were calculated using density functional theory at the level of B3-LYP/6-31(d,p) with the correlation of Grimme's D2 dispersion term using CrystalExploer software[44,45]. For the interaction energy calculations, molecules located within 3.8 Å of the central molecule were considered, and disordered *tert*-butyl groups with minor occupancy were deleted by editing the cif file. The intermolecular interaction energy is the summation of polarization, electrostatic, repulsion, and dispersion energies with scale factor coefficients of 1.057, 0.740, 0.871, and 0.618, respectively[46]. The energy framework was drawn by a solid line, and the line width reflects the relative value of the intermolecular interaction energies, with cut-off energy of 5.0 kJ mol$^{-1}$.

**Measurement of mechanical properties.** The actuation behavior of a crystal irradiated by UV light (365 nm) with a UV-LED (8332 A, CCS Inc.) was observed using a microscope camera (VW-6000, Keyence), and then analyzed by Tracker[47]. Load–displacement curves of enol-(S)-**1** crystals glued on a glass plate were measured at ambient condition (20 °C) with a universal material testing machine (Tensilon RTG-1210, A&D Co. Ltd.) at a moving speed of the jig of 0.2 mm min$^{-1}$. Each measurement was monitored using a camera (A1microscope, MixMart). Force measurements were performed with the same testing machine in the creep mode, and light irradiation at 365 nm was performed using the UV-LED (8332 A, CCS Inc.) equipped with a φ12-mm lens. Force measurement was conducted as the following procedure: the jig slowly approached the crystal before light irradiation and stopped at contact with the crystal surface, which was confirmed by detecting a small initial load (approximately 0.2 mN), and the blocking force was monitored under and after UV irradiation. The light intensity was calibrated using a UV radiometer (UVR-300, Topcon Technohouse Co.).

**Measurement of surface temperature.** Surface temperature changes upon UV irradiation and after removal were measured using infrared (IR) thermography (FSV-200, Apiste). Light irradiation onto the enol-(S)-**1** crystal was also conducted by the UV-LED (8332 A, CCS Inc.) with a φ12-mm lens.

**FT-IR measurement.** A spectrometer FT/IR-6600 (JASCO, Japan) equipped with an ATR module (ATR PRO ONE with germanium-prism) was used for FT-IR measurements. The measurements of the enol-(S)-**1** powder were performed at 20 °C, and light irradiation of the enol-(S)-**1** powder was also conducted using the UV-LED (8332A, CCS Inc.) with a φ12-mm lens. For theoretical calculation of the IR spectrum, an independent molecule was extracted from the crystal structure at 20 °C, and then, after deleting minor disorder of the *tert*-butyl substituent, the

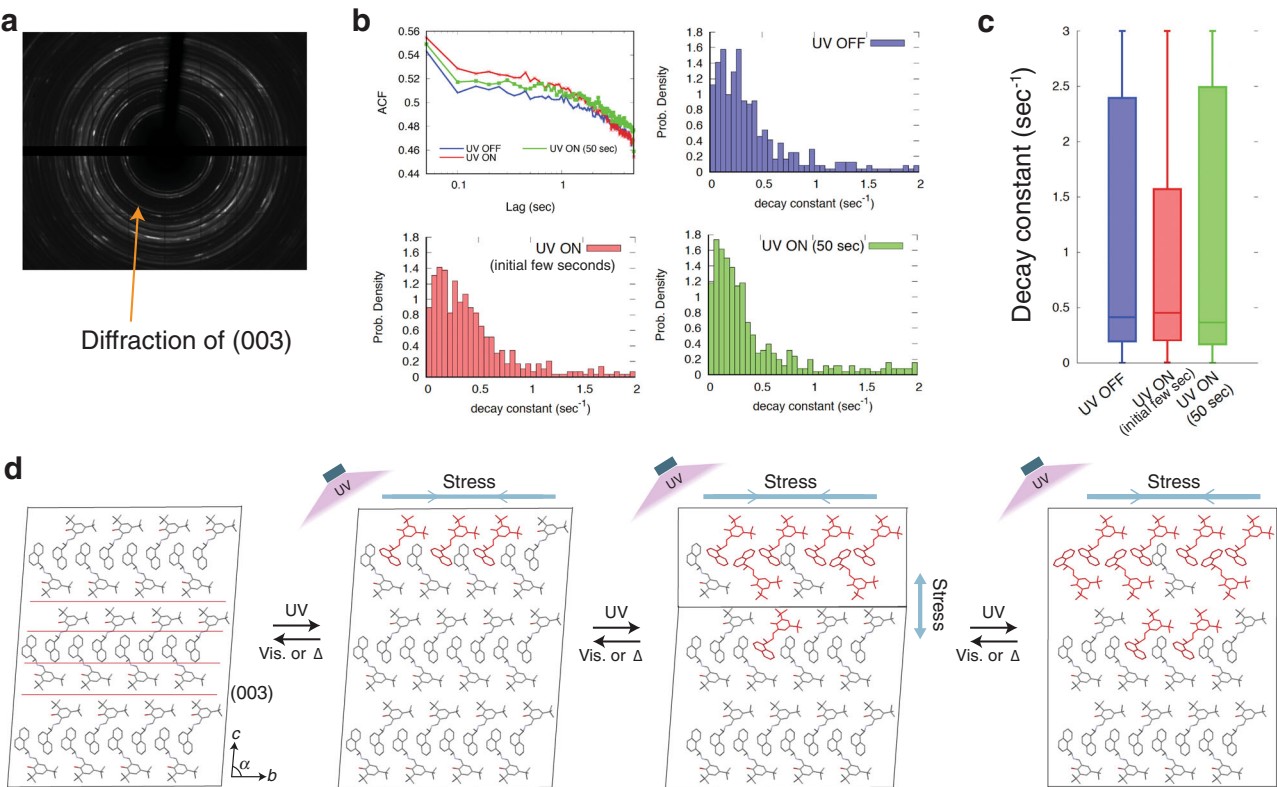

**Fig. 6 Crystal structure dynamics and proposed mechanism of superelasticity. a** X-ray diffraction image of enol-(*S*)-**1** powder. **b** Mean autocorrelation function (ACF) and probability density of ACF decay constants before light irradiation, and upon initial and prolonged irradiation. **c** Boxplot of the decay constants. The boxes show the median and first/third quartiles. **d** Manifestation mechanism of superelasticity based on crystal structure dynamics and generated stress upon light irradiation.

molecular geometry was optimized at the theoretical level of B3-LYP/6-31 G(d,p) using Gaussian[48]. The *trans*-keto form was prepared by editing the geometry of enol-(*S*)-**1** and optimized using the same procedure as that used for enol-(*S*)-**1**. Finally, the IR spectra were calculated based on these optimized molecules.

**Polarized microscopy**. Phase boundary was observed under a polarized microscope (DM LM/P 11888500, Leica), recorded by a digital camera (WRAYCAM-NF300, Wraymer). A crystal sample was fixed to the glass plate by glue to prevent actuation as much as possible. Light irradiation was conducted using the UV-LED (8332A, CCS Inc.).

**Numerical analysis of crystal deformation**. The deformation of the enol-(*S*)-**1** crystal was simulated by FEA using Ansys software ver. 2021 R1 (ANSYS, Inc.) on a desktop computer equipped with AMD Ryzen 9 (16 cores, 64 GB). Before the simulation, the material geometry was created using ANSYS SpaceClaim. The material properties were defined in the Ansys workbench. The static analysis of deformation was performed in Ansys mechanical, where the effects of photo-isomerization and PtPT were incorporated as thermal contraction and piezo effects, respectively. These parameters are shown in Supplementary Fig. 10. Three-dimensional visualization and hyperparameter optimization were performed using matplotlib and scipy packages based on Python.

**Diffracted X-ray blinking (DXB)**. The DXB measurements were performed using a laboratory X-ray source (MicroMax-007 HF: Cu anode, wavelength $\lambda = 1.54$ Å, 40 kV, 30 mA). Time-resolved diffraction images were recorded using a 2D photon-counting detector (Pilatus 200 K-A, Dectris, Switzerland) with a time resolution of 50 ms. The diffraction intensity of a specific $2\theta$ value in each pixel was analyzed. The time-resolved intensity at each pixel was analyzed using the ACF,

$$I(\kappa) = \frac{I(t)I(t+\kappa)}{I(t)^2} \quad (5)$$

where $I(t)$ is the diffraction intensity, brackets <> indicate the time-averaged values, and $\kappa$ is the lag time (or interval). The ACF can be computed by changing $\kappa$, and then fitted to an exponential curve by $ACF(t) = A \exp(-\Gamma t) + y$, where $A$ is the amplitude, $y$ is the conversational value, and $\Gamma$ is the decay constant. Through this fitting, the decay constant at each pixel was obtained. We chose decay constants to satisfy the following conditions: (I) $0 < y$, $0 < A$ and $0 < \Gamma$, and (II) residual values

between the fitted and actual ACF curves of less than 1.0. These calculations were performed for all pixels. The distribution of decay constants of a specific diffraction ring was visualized using a histogram and boxplot to estimate the dynamic behavior of the crystal structure. Light irradiation was conducted by the UV-LED (8332A, CCS Inc.) with a φ12-mm lens.

## Data availability
The datasets generated and/or analyzed during the current study are available from the corresponding author upon reasonable request.

## Code availability
The codes that support the findings of this study are available from the corresponding author upon reasonable request.

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

## Acknowledgements

This study was financially supported by the JSPS Grant-in-Aid (16K12918, 17H03107, 19K23638, 20H04677, 20H04660, 20H04696, and 21K14466) and Waseda University Grant for Special Research Projects (2019C-646, 2020C-530, and 2020E-076, 2021C-404). This work was also partially supported by the Cabinet Office, Government of Japan, Cross-ministerial Moonshot Agriculture, Forestry and Fisheries Research and Development Program "Technologies for Smart Bio-industry and Agriculture" (BRAIN).

## Author contributions

T.T. managed the project and wrote the paper. K.I. performed the movie analysis. D.T. calculated the energy framework. H.S. and K.N. assisted with FEA. M.K. and Y.C.S. performed the DXB measurement and analysis. T.T. performed other experiments and all simulations. H.K. and T.A. helped supervise the project. All the authors reviewed the paper and contributed to useful discussions.

## Competing interests

The authors declare no competing interests.
