## [Peer Review File · Communications Chemistry]

Reviewers' comments:

Reviewer #1 (Remarks to the Author):

Photoactuation in molecular crystals of optically switchable molecules has become a busy area of investigation. The present study of phototriggered phase transformation brings something new, a photoactuated superelasticity. The work is admirably comprehensive bringing together kinetic models, mechanical measurements and simulations, and unusual methods of analysis (X-ray blinking). There are a few places where clarity would be helpful, but overall this work is the the very top tier of papers in this class.

In the abstract, the authors argue that superelasticity and actuation are mixed up conceptually. However, by the definitions given in the abstraction and the continued use of the terms (actuation 4x and superelasticity 3x) I don't really appreciate the distinction being made. The reader should leave the abstract knowing this much. The clarifying phrase comes on page 5, whether force is input or out. This needs to be said here too, first.

The effect of super or pseudo elasticity (incidentally misleading/confusing names) are not so much accompanied by phase transitions but are the manifestations of phase transitions (martensitic). It is not like two things happening at the same but rather a deformation affected by the colligative transformation.

Minor comments:

Abstract. "conceptual gap between the passive and active behaviors"
I am confused by the phrase "diverting the twist model" on page 5.

In Figure 3, I find it very difficult to really "see" how the crystal shape is changing, which is real, what are artifacts. The bound crystals look like they are being translated.

Torsion is twisting due to torque. Thus, when the authors say "torsional shape", I know precisely what they mean, but it is a strange usage. I would say "twisted shape" instead.

On page 15, change "twisted bending" to "twist-bent deformation".

FEA on page 15 needs to be defined as finite element analysis.

I don't understand how the X-ray blinking autocorrelation function "clarifies the dynamic changes in the crystal structure". Page 16. "The decay constant was distributed..." ??

Reviewer #2 (Remarks to the Author):

This paper submitted by Taniguchi et al. reports the underlying mechanism and photomechanical response of molecular crystals based on acylhydrazone derivatives. Crystals of this acylhydrazone derivatives can EZ isomerization in the crystalline state. The UV light irradiation leads to bending phenomena or twisting of the crystal. This behavior in crystals is not unique, and the field of photo-induced crystal responses is overripe. The phenomenon is so common that new examples appear weekly. However, this paper presents an interesting system that will be attractive to readers in and out of the photomechanical crystal community.

This paper misses elaborating the molecular mechanism for why the transformations occur. The authors have not clarified molecular mechanism for phase transformation nor the correlation between transformation and the amount of isomerized molecules.

Reviewer #3 (Remarks to the Author):

Takuya et al. report the observation and characterization of superelasticity of a photo-activated response in the molecular crystal of the enol-(S)-1 compound. The superelasticity is interpreted as the result of stepwise twisted actuation due to two effects, photoisomerization and photo-triggered phase transition. The result is interesting, and the interpretation is insightful. The study could have gone a little further, as suggested below.

Comments/suggestions

1. This reviewer considers the definition/difference between superelasticity and actuation (and discussion of them) is less meaningful because both the behaviors are indeed the "passive" response to the external stimuli, whether by light or heat (actuation) or by mechanical force. In the former, light/heat may drive conformational changes, which in turn drive the phase transition. In the latter, the mechanical force might drive the transition directly.
2. I suggest the authors perform a kinetics study of the enol-(S)-1 compound photolysis in solution under various light intensities and light pulse widths to establish the baseline limit of the photolysis rate. The 1% conversion in crystal seems to be extremely low. Could it be due to the lattice restriction?
3. The layer and stepwise model are attractive. By this model, the authors appear to rule out the possibility of the "synchrony" model where all molecules undergo the phase transition simultaneously. What would happen if the photolysis conversion rate is 100%? Do you expect to observe the "synchrony" of the transition?
4. I'm very curious if the authors have considered recording the birefringence of the transition as a function of, for example, UV light intensity over the course of time and to compare with the time trace of the simulated parameters, such as displacement and torsion angles. Such a birefringence experiment and data analysis software for analysis of phase transition behaviors in crystal have recently been reported in the literature. One of the advantages of using birefringence over other methods to monitor the phase transition is that it can track the phase changes at the pixel level. Thus one can monitor and study micron/nanocrystals and phase transition behaviors at the pixel level, far beyond the "bulk" level as reported in this paper. With micron/nanocrystals, one might explain the transition in terms of the synchrony" model instead of the "layer and stepwise" model, which might be the result of non-uniform excitation of very large crystal by UV.
5. The authors might want to consider repeating the experiments under low temperatures, such as -20 °C or even lower, to eliminate/rule out local thermal "melting" as a cause of the heterogeneous transitions.
6. Show data of more than just one crystal in order to see the impact of crystal heterogeneity on the phase transition.

Point-by-point responses to reviewers' comments

Comments by Reviewer #1:

Photoactuation in molecular crystals of optically switchable molecules has become a busy area of investigation. The present study of phototriggered phase transformation brings something new, a photoactuated superelasticity . The work is admirably comprehensive bringing together kinetic models, mechanical measurements and simulations, and unusual methods of analysis (X-ray blinking). There are a few places where clarity would be helpful, but overall this work is the the very top tier of papers in this class.

In the abstract, the authors argue that superelasticity and actuation are mixed up conceptually. However, by the definitions given in the abstraction and the continued use of the terms (actuation 4x and superelasticity 3x) I don't really appreciate the distinction being made. The reader should leave the abstract knowing this much. The clarifying phrase comes on page 5, whether force is input or out. This needs to be said here too, first.

The effect of super or pseudo elasticity (incidentally misleading/confusing names) are not so much accompanied by phase transitions but are the manifestations of phase transitions (martensitic). It is not like two things happening at the same but rather a deformation affected by the colligative transformation.

Minor comments:

Abstract. "conceptual gap between the passive and active behaviors"

I an confused by the phrase "diverting the twist model" on page 5.

In Figure 3, I find it very difficult to really "see" how the crystal shape is changing, which is real, what are artifacts. The bound crystals look like they are being translated.

Torsion is twisting due to torque. Thus, when the authors say "torsional shape", I know precisely what they mean, but it is a strange usage. I would say "twisted shape" instead.

On page 15, change "twisted bending" to "twist-bent deformation".

FEA on page 15 needs to be defined as finite element analysis.

I don't understand how the X-ray blinking autocorrelation function "clarifies the dynamic changes in the crystal structure". Page 16. "The decay constant was distributed..." ??

Author response to reviewer #1:

Thank you for the good evaluation. We have revised the manuscript according to the comments.

Comment 1: In the abstract, the authors argue that superelasticity and actuation are mixed up conceptually. However, by the definitions given in the abstraction and the continued use of the terms (actuation 4x and superelasticity 3x) I don't really appreciate the distinction being made. The reader should leave the abstract knowing this much. The clarifying phrase comes on page 5, whether force is input or out. This needs to be said here too, first.

Reply 1: Thank you for the advisory comment. We have revised abstract and introduction according to the comment. For example, in abstract "there has been a conceptual gap between passive and active behaviors" was changed to "there is a phenomenological difference whether force is input or output" for the explanation of superelasticity and actuation.

Comment 2: The effect of super or pseudo elasticity (incidentally misleading/confusing names) are not so much accompanied by phase transitions but are the manifestations of phase transitions (martensitic). It is not like two things happening at the same but rather a deformation affected by the colligative transformation.

Reply 2: Thank you for the advisory comment. We have deleted "superelasticity is accompanied by phase transition" from the whole manuscript, and revised to clarify that superelasticity is a deformation or manifestation of phase transitions.

Comment 3: Abstract. "conceptual gap between the passive and active behaviors"

Reply 3: We have changed "conceptual gap between the passive and active behavior" to "phenomenological difference whether force is input or output" (on page 3).

Comment 4: I am confused by the phrase "diverting the twist model" on page 5.

Reply 4: According to the comment, "by diverting the twist model ~" was deleted.

Revised sentence is “The twisted actuation has been, in some cases, analyzed successfully by analytical models³⁴.” (on page 5)

Comment 5: In Figure 3, I find it very difficult to really “see” how the crystal shape is changing, which is real, what are artifacts. The bound crystals look like they are being translated.

Reply 5: We observed the photoinduced motion of a crystal fixed with a glass plate or needle from two different views. Figure 3a shows a side view, at which we can see bending clearly but twist cannot be evaluated. Figure 3c,f show a cross-section view, at which we cannot see bending curvature but can analyze both twist and bending quantitatively by torsion angle and tip displacement, respectively, based on the definition in Figure 3b. For clarity, we added the description of fixation in the figure panel and the caption. (on page 41)

Comment 6: Torsion is twisting due to torque. Thus, when the authors say “torsional shape”, I know precisely what they mean, but it is a strange usage. I would say “twisted shape” instead.

Reply 6: Thank you for the suggestion, and according to the comment, we changed “torsional shape” to “twisted shape” in the manuscript.

Comment 7: On page 15, change “twisted bending” to “twist-bent deformation”.

Reply 7: According to the comment, we changed “twisted bending” to “twist-bent deformation”.

Comment 8: FEA on page 15 needs to be defined as finite element analysis.

Reply 8: According to the comment, we changed “FEA” to “finite element analysis (FEA)” to define the abbreviation.

Comment 9: I don’t understand how the X-ray blinking autocorrelation function “clarifies the dynamic changes in the crystal structure”. Page 16. “The decay constant was distributed...” ??

Reply 9: We understand the concern because diffracted X-ray blinking (DXB) is a unique measurement method developed by Kuramochi and Sasaki, co-authors of this manuscript. For further explanation, we added Supplementary Note 3, where we mention in more detail how is data processed, and what is distribution of decay constants.

Comments by Reviewer #2:

This paper submitted by Taniguchi et al. reports the underlying mechanism and photomechanical response of molecular crystals based on acylhydrazone derivatives. Crystals of this acylhydrazone derivatives can EZ isomerization in the crystalline state. The UV light irradiation leads to bending phenomena or twisting of the crystal. This behavior in crystals is not unique, and the field of photo-induced crystal responses is overripe. The phenomenon is so common that new examples appear weekly. However, this paper presents an interesting system that will be attractive to readers in and out of the photomechanical crystal community.

This paper misses elaborating the molecular mechanism for why the transformations occur. The authors have not clarified molecular mechanism for phase transformation nor the correlation between transformation and the amount of isomerized molecules.

Author response to reviewer #2:

Thank you for reviewing the manuscript.

Comments by Reviewer #3:

Takuya et al. report the observation and characterization of superelasticity of a photo-activated response in the molecular crystal of the enol-(S)-1 compound. The superelasticity is interpreted as the result of stepwise twisted actuation due to two effects, photoisomerization and photo-triggered phase transition. The result is interesting, and the interpretation is insightful. The study could have gone a little further, as suggested below.

Comments/suggestions

1. This reviewer considers the definition/difference between superelasticity and actuation (and discussion of them) is less meaningful because both the behaviors are indeed the “passive” response to the external stimuli, whether by light or heat (actuation) or by mechanical force. In the former, light/heat may drive conformational changes, which in turn drive the phase transition. In the latter, the mechanical force might drive the transition directly.
2. I suggest the authors perform a kinetics study of the enol-(S)-1 compound photolysis in solution under various light intensities and light pulse widths to establish the baseline limit of the photolysis rate. The 1% conversion in crystal seems to be extremely low.

Could it be due to the lattice restriction?

3. The layer and stepwise model are attractive. By this model, the authors appear to rule out the possibility of the “synchrony” model where all molecules undergo the phase transition simultaneously. What would happen if the photolysis conversion rate is 100%? Do you expect to observe the “synchrony” of the transition?

4. I’m very curious if the authors have considered recording the birefringence of the transition as a function of, for example, UV light intensity over the course of time and to compare with the time trace of the simulated parameters, such as displacement and torsion angles. Such a birefringence experiment and data analysis software for analysis of phase transition behaviors in crystal have recently been reported in the literature. One of the advantages of using birefringence over other methods to monitor the phase transition is that it can track the phase changes at the pixel level. Thus one can monitor and study micron/nanocrystals and phase transition behaviors at the pixel level, far beyond the “bulk” level as reported in this paper. With micron/nanocrystals, one might explain the transition in terms of the synchrony” model instead of the “layer and stepwise” model, which might be the result of non-uniform excitation of very large crystal by UV.

5. The authors might want to consider repeating the experiments under low temperatures, such as -20 °C or even lower, to eliminate/rule out local thermal “melting” as a cause of the heterogeneous transitions.

6. Show data of more than just one crystal in order to see the impact of crystal heterogeneity on the phase transition.

Author response to reviewer #3:

Thank you for the good evaluation and important comments for our paper. We have revised the manuscript according to the comments.

Comment 1: This reviewer considers the definition/difference between superelasticity and actuation (and discussion of them) is less meaningful because both the behaviors are indeed the “passive” response to the external stimuli, whether by light or heat (actuation) or by mechanical force. In the former, light/heat may drive conformational changes, which in turn drive the phase transition. In the latter, the mechanical force might drive the transition directly.

Reply 1: Thank you for the advisory comment. We have deleted the distinction of active/passive, and changed to represent that both superelasticity and actuation are deformations, and the difference between them relied on whether force is an input or

output. We have modified the whole manuscript to clarify this point. For example, in abstract (on page 3)

“there has been a conceptual gap between passive and active behaviors” was changed to “there is a phenomenological difference whether force is input or output”.

Comment 2: I suggest the authors perform a kinetics study of the enol-(S)-1 compound photolysis in solution under various light intensities and light pulse widths to establish the baseline limit of the photolysis rate. The 1% conversion in crystal seems to be extremely low. Could it be due to the lattice restriction?

Reply 2: Salicylideneamines, the family name of enol-(S)-1 in this research, generally exhibit significant differences in chromic properties in solution and in the solid state. Trans-isomer is formed by proton transfer and pedal movement from enol isomer, resulting in photochromism. In solution, the trans-form returns to the enol-form very fast, and the color change cannot be seen. On the other hand, the solid state of salicylideneamines can exhibit photochromism owing to limited free space, which makes the conversion from trans-keto to enol form slow. In this study, we estimated that the conversion ratio of photoproduct was 5% at steady state, and this value is consistent with the previous study that the ratio of trans-keto was not more than 10% because it was insufficient for an X-ray structure determination even as disorder. This is also comparable with the conversion ratio of other photochromic crystals in some literature. To clarify this, we have added the following sentence to the text.

“This value is consistent with the previous research that conversion ratio of photoisomerization was insufficient for an X-ray structure determination³⁵, and comparable with conversion ratio of similar photochromic crystals^{24,34}.” (on page 14)

Comment 3: The layer and stepwise model are attractive. By this model, the authors appear to rule out the possibility of the “synchrony” model where all molecules undergo the phase transition simultaneously. What would happen if the photolysis conversion rate is 100%? Do you expect to observe the “synchrony” of the transition?

Reply 3: First, let us clarify that the proposed model is composed of two phenomena: photoisomerization and PtPT. Photoisomerization is the molecular conversion from enol to trans-keto form, and PtPT is the phase transition accompanied by the change of molecular conformation and arrangement of mainly enol-form. Here, we consider two imaginary situations.

1. The case that all molecules undergo the phase transition (PtPT) simultaneously

In this case, twisted shape does not appear because it occurs during the progression of

phase transition.

2. The case that photolysis conversion rate is 100%

In this case, all molecules are trans-keto form, and thus the crystal must be a new phase which we have not met.

Such situations are not consistent with our observations and analyses. In addition, we experimentally observed the progression of phase boundary upon UV irradiation and after the cease, under a polarized microscope (related to reply 4). Therefore, we think that the layer and stepwise model should be more suitable than synchrony model.

Comment 4: I'm very curious if the authors have considered recording the birefringence of the transition as a function of, for example, UV light intensity over the course of time and to compare with the time trace of the simulated parameters, such as displacement and torsion angles. Such a birefringence experiment and data analysis software for analysis of phase transition behaviors in crystal have recently been reported in the literature. One of the advantages of using birefringence over other methods to monitor the phase transition is that it can track the phase changes at the pixel level. Thus one can monitor and study micron/nanocrystals and phase transition behaviors at the pixel level, far beyond the "bulk" level as reported in this paper. With micron/nanocrystals, one might explain the transition in terms of the synchrony" model instead of the "layer and stepwise" model, which might be the result of non-uniform excitation of very large crystal by UV.

Reply 4: Thank you for the suggestive comment. We have additionally done the birefringence observation of the crystal under and after UV light irradiation, and we noticed that phase boundary progressed during the phase transformation. We added the results as Supplementary Figure 16 and Supplementary Movie 5. Although the birefringence change is very small, the phase boundary appears almost parallel to (001) plane upon the light irradiation to (001) face, and the boundary returned reversibly after the light was turned off. This observation also supported layer and stepwise model rather than synchrony model. The time scale was almost consistent with the result of FEA-based simulation.

Although we haven't considered size dependence on this behavior, we investigated the anisotropy of the phase transition by light irradiation to different faces. Irradiation to (010) face caused phase boundary partially, and the irradiation to (100) face did not cause a noticeable transition. We added the result as Supplementary Figure 17. This observation suggests that light irradiation to (001) face is the most effective to induce PtPT.

To mention the additional result, we added following sentences:

“Indeed, phase boundary and slight shear deformation were observed during PtPT and reverse transition under a polarized microscope (Supplementary Figure 16 and Supplementary Movie 5). The phase boundary was generated parallel to (001) face, to which UV irradiation was conducted. This appearance of phase boundary is common with superelasticity, and the shear deformation of enol-(S)-1 crystal was much smaller than those of other superelastic crystals^{3,4,5}. Furthermore, the anisotropy of PtPT expression was investigated (Supplementary Figure 17). UV irradiation to (010) face led to the partial generation of phase boundary, while light irradiation to (100) face did not cause such boundary. This observation suggests that light irradiation to (001) face is the most effective to induce PtPT probably due to the anisotropy of crystal structure, photo-reactivity, and large area of the face. In all observations, we did not detect any local melt, which would affect the PtPT and deformation behavior (Supplementary Figure 17).” (on page 20)

Comment 5: The authors might want to consider repeating the experiments under low temperatures, such as -20 °C or even lower, to eliminate/rule out local thermal “melting” as a cause of the heterogeneous transitions.

Reply 5: Related to the above reply 4, we checked if the crystal partially melts upon UV irradiation using polarized microscopy. We have not detected any melting even at the highest intensity of UV light (360 mW/cm²). We added this result in Supplementary Figure 17. Thus, we should consider the mechanical motion as the event at a totally crystal state. Of course, we have an interest in temperature dependence (both low and high temperatures), but such experiments change the parameters of material mechanics such as Young’s modulus. So, we think it's better to do it in future work.

Comment 6: Show data of more than just one crystal in order to see the impact of crystal heterogeneity on the phase transition.

Reply 6: According to the comment, we have done additional observations and analyses. We have added the result as Supplementary Figure 15, and confirmed that the layer and stepwise model is consistent with the deformation of other crystals of the same compound. Although there is some misfit, the simulated torsion angle fits well with the observations. Displacements also outperformed classical Stoney’s model.

REVIEWERS' COMMENTS:

Reviewer #1 (Remarks to the Author):

Generally speaking, the reviewers were of the opinion that the authors brought something new to the study of photomechanical systems, in spite of their stated apprehension of the explosive growth of this subject.

The specific comments were addressed openly and honestly.

I continue to support publication.

Reviewer #3 (Remarks to the Author):

No further comments. This reviewer is satisfied with the current revision.

Comments by Reviewer #1:

Generally speaking, the reviewers were of the opinion that the authors brought something new to the study of photomechanical systems, in spite of their stated apprehension of the explosive growth of this subject.

The specific comments were addressed openly and honestly.

I continue to support publication.

Author response to reviewer #1:

Thank you for the review of our manuscript, and the recommendation of publication.

Comments by Reviewer #3:

No further comments. This reviewer is satisfied with the current revision.

Author response to reviewer #3:

Thank you for the review of our manuscript, and the recommendation of publication.